# Integrating evolutionary and regulatory information with multispecies approach implicates genes and pathways in obsessive-compulsive disorder

Hyun Ji Noh et al.#

Obsessive-compulsive disorder is a severe psychiatric disorder linked to abnormalities in glutamate signaling and the cortico-striatal circuit. We sequenced coding and regulatory elements for 608 genes potentially involved in obsessive-compulsive disorder in human, dog, and mouse. Using a new method that prioritizes likely functional variants, we compared 592 cases to 560 controls and found four strongly associated genes, validated in a larger cohort. NRXN1 and HTR2A are enriched for coding variants altering postsynaptic protein-binding domains. CTTNBP2 (synapse maintenance) and REEP3 (vesicle trafficking) are enriched for regulatory variants, of which at least six (35%) alter transcription factor-DNA binding in neuroblastoma cells. NRXN1 achieves genome-wide significance ($p = 6.37 \times 10^{-11}$) when we include 33,370 population-matched controls. Our findings suggest synaptic adhesion as a key component in compulsive behaviors, and show that targeted sequencing plus functional annotation can identify potentially causative variants, even when genomic data are limited.

#A full list of authors and their affliations appears at the end of the paper

Obsessive-compulsive disorder (OCD) is a highly heritable ($h^2 = 0.27$–$0.65$)[1], debilitating neuropsychiatric disorder characterized by intrusive thoughts and time-consuming repetitive behaviors. Over 80 million people worldwide are estimated to suffer from OCD, and most do not find relief with available therapeutics[1], underscoring the urgency to better understand the underlying biology. Genome-wide association studies (GWAS) implicate glutamate signaling and synaptic proteins[2, 3], but specific genes and variants have not been validated. Isolating and characterizing such genes are important for understanding the biology and developing treatments for this devastating disease.

In mouse, genetically engineered lines have causally implicated the cortico-striatal neural pathway in compulsive behavior. Mice with a deletion of *Sapap3* exhibit self-mutilating compulsive grooming and dysfunctional cortico-striatal synaptic transmission, with abnormally high activity of medium spiny neurons (MSNs) in the striatum. Resulting compulsive grooming is ameliorated by selective serotonin reuptake inhibitor (SSRI), a first-line medication for OCD[4]. Similarly, chronic optogenetic stimulation of the cortico-striatal pathway in normal mice leads to compulsive grooming accompanied by sustained increases in MSN activity[5]. Thus, excessive striatal activity, likely due to diminished inhibitory drive in MSN microcircuitry, is a key component of compulsive grooming. The brain region disrupted in this mouse model is also implicated by imaging studies in human OCD[6].

Pet dogs are a natural model for OCD amenable to genome-wide mapping due to their unique population structure[7]. Canine compulsive disorder (canine CD) closely parallels OCD, with equivalent clinical metrics, including compulsive extensions of normal behaviors, typical onset at early social maturity, roughly a 50% rate of response to SSRIs, high heritability, and polygenic architecture[8]. Through GWAS and targeted sequencing in dog breeds with exceptionally high rates of canine CD, we associated genes involved in synaptic functioning and adhesion with CD, including neural cadherin (*CDH2*), catenin alpha2 (*CTNNA2*), ataxin-1 (*ATXN1*), and plasma glutamate carboxypeptidase (*PGCP*)[8, 9].

Human genetic studies of related disorders, such as autism spectrum disorders (ASD), suggest additional genes. Both ASD and OCD are characterized by repetitive behaviors, and high comorbidity suggests a shared genetic basis[6]. Genome-wide studies searching for de novo and inherited risk variants have confidently associated hundreds of genes with ASD; this set may be enriched for genes involved in OCD[10].

Focusing on genes implicated by model organisms and related disorders could find variants underlying OCD risk, even with smaller sample sizes. Researchers, particularly in psychiatric genetics, are wary of "candidate gene" approaches, which often failed to replicate[11]. Closer examination of past studies suggests this approach is powerful and reliable when the set of genes tested is large, and the association is driven by rare variation[11]. A study testing 2000 candidate genes for association with diabetic retinopathy identified 25 genes, at least 11 of which achieved genome-wide significance in a GWAS of type 2 diabetes, a related disorder[12, 13]. A targeted-sequencing study of ASD, with 78 genes, identified four genes with recurrent, rare deleterious mutations; these four genes are also implicated by whole-exome sequencing studies[14]. Candidate gene studies also replicated associations to rare variants in *APP*, *PSEN1*, and *PSEN2* for Alzheimer's disease[15], *PCSK9* for low-density lipoprotein–cholesterol level[16], and copy-number variants for autism and schizophrenia[10].

Detecting associations driven by rare variants requires sequencing data, which captures nearly all variants. Although whole-genome sequencing studies of complex diseases are still prohibitively expensive, it is feasible to target a subset of the genome. Sequencing also facilitates identification of causal variants, accelerating discovery of new therapeutic avenues[17, 18]. For example, finding functional, rare variants in *PCSK9* led to new therapies for hypercholesterolemia[19]. One approach is to target predominantly coding regions (whole-exome sequencing). Although successful in finding causal variants for rare diseases[20], this approach misses the majority of disease-associated variants predicted to be regulatory[21]. A targeted-sequencing approach that captures both the regulatory and coding variation of a large set of candidate genes offers many advantages of whole-genome sequencing, and is feasible when cohort size and resources are limited.

Here we report a new strategy that overcomes limitations of less comprehensive candidate gene studies and exome-only approaches, and identifies functional variants associated with increased risk of OCD. We start by compiling a large set of 608 genes (~3% of human genes) using studies of compulsive behavior in dogs and mice, and studies of ASD and OCD in humans. By focusing on this subset of genes, targeting both coding and regulatory regions, and applying a new statistical method that incorporates regulatory and evolutionary information, we identify four associated genes, including *NRXN1*, the first genome-wide-significant association reported for OCD.

## Results

**Targeted-sequencing design**. We compiled a list of 608 genes using three strategies (65 were implicated more than once) (Supplementary Table 1 and Supplementary Methods):

(1) 263 "model-organism genes", including 56 genes associated in canine CD GWAS and 222 genes implicated in murine-compulsive grooming.
(2) 196 "ASD genes" from SFARI database (https://gene.sfari.org/) as of 2009.
(3) 216 "human candidate genes" from small-scale OCD candidate gene studies (56 genes), family-based linkage studies of OCD (91 genes), and by other neuropsychiatric disorders (69 genes).

We targeted coding regions and 82,723 evolutionarily constrained elements in and around these genes, totaling 13.2 Mb (58 bp–16 kb size range, median size 237 bp), 34% noncoding[22].

**Variant detection**. We sequenced 592 European ancestry DSM-IV OCD cases and 560 ancestry-matched controls using pooled sequencing, with 16 samples per bar-coded pool (37 "case" pools; 35 "control" pools). Overall, 95% of target regions were sequenced at $>30\times$ read depth per pool (median $112\times$; ~$7\times$ per individual; Supplementary Fig. 1), sufficient to identify variants occurring in just one individual, assuming 0.5–1% per base machine error rate.

We called 124,541 single nucleotide polymorphisms (SNPs) using Syzygy (84,216)[17] and SNVer (81,829)[23]. For primary analyses, we focused on 41,504 "high-confidence" SNPs detected by both, with highly correlated allele frequencies (AF) (Pearson's $\rho = 0.999$, $p < 2.2 \times 10^{-16}$; Supplementary Fig. 2). We see no significant difference between case and control pools, indicating no bias in variant detection.

**Variant annotation**. We used three annotations shown to be enriched for disease-associated variation to identify likely functional variants in our targeted regions: coding, evolutionary conserved, and/or DNase1 hypersensitivity site (DHS)[21, 24–27]. We annotated 67% (27,626) of high-confidence variants, with 16% coding (49% of those were non-synonymous), 36% DHS,

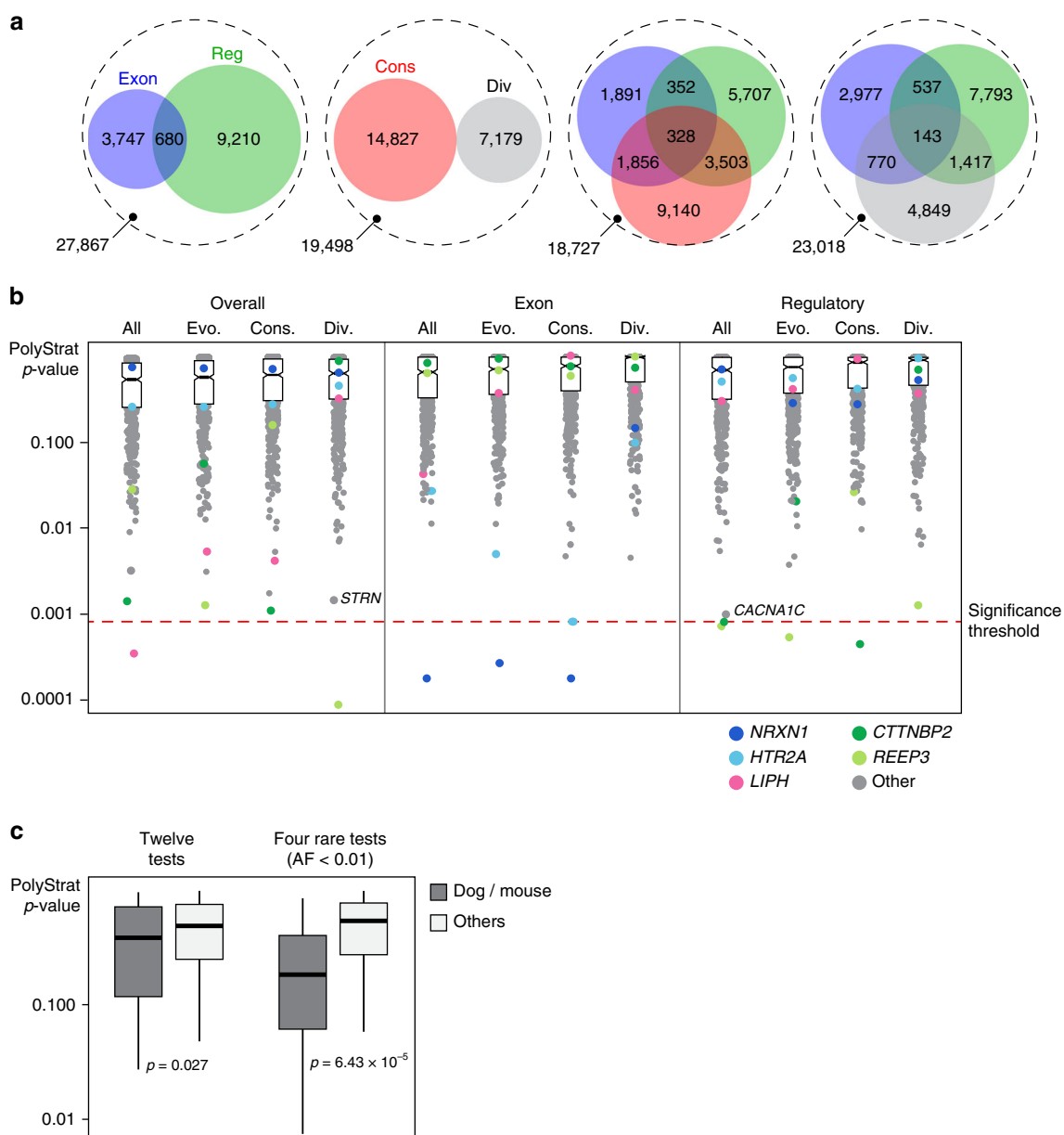

**Fig. 1** PolyStrat analysis of pooled-targeted-sequencing data. **a** Venn diagrams showing the number of SNPs annotated as functional and/or conserved by PolyStrat. Each of the four *dashed circles* represents the 41,504 total high-confidence SNPs detected. Within each *circle*, SNPs are stratified by their annotations. Each *colorful interior circle* represents SNPs annotated as exonic (*blue*), regulatory (*green*), conserved (*red*), or diverged (*gray*) bases. SNPs with multiple annotations are represented by *circle overlaps*, and SNPs without any of the included annotations are within the *white space* of the *dashed outer circle*. **b** PolyStrat *p*-values for 608 genes (*circles*) stratified by the 16 (12 shown) annotation categories tested show that just five genes (*NRXN1, HTR2A, LIPH, CTTNBP2,* and *REEP3*) have *p*-values below the experiment-wide significance threshold after correction for multiple testing (*red dashed line*). Two moderately associated, OCD-relevant genes discussed in the text are also noted (*STRN* and *CACNA1C*). "Evo" (=evolutionary) are SNPs either conserved ("Cons") or divergent ("Div"). The vast majority of genes tested fail to exceed the significance threshold, with the median *p*-value for each category shown as a *dark black line* separating two *boxes* representing the 25–75% quantile. *Notch* in *boxes* shows the 95% confidence interval around median. **c** *p*-values for the five genes robustly implicated in animal models of OCD are significantly lower than *p*-values for the rest of the genes in our sequencing set (603 genes), and this difference increases when just rare variants are tested. The *solid horizontal line* shows median *p*-value, the *boxed area* the 25–75% quantiles, and the *vertical black lines* extend from the minimum to maximum *p*-values observed. AF, allele frequency

and 80% evolutionary conserved or divergent (Fig. 1a). We measured evolutionary constraint using mammalian GERP++ scores[27]; scores >2 were "conserved" and scores <−2 were "divergent".

**Gene-based burden analysis.** To identify genes with a significant load of non-reference alleles in OCD cases, relative to controls, we developed PolyStrat, a one-sided gene-based burden test that

controls for gene length (Supplementary Fig. 3a) and incorporates variant annotation. We used four variant categories: (i) all (Overall), (ii) coding (Exon), (iii) regulatory (variants in DHS), and (iv) rare (1000 Genomes Project[28] AF < 0.01). Each category is further stratified by evolutionary status: (i) all detected variants; (ii) slow-evolving conserved (Cons); (iii) fast-evolving divergent (Div); and (iv) evolutionary (Evo). "Evo" is the subset of "all" variants annotated as either "conserved" or "divergent". In total,

**Table 1 Five genes with significant variant burden in OCD cases in pooled sequencing data**

| Genes | Description | Total SNPs in PolyStrat | Case-abundant SNPs in PolyStrat | PolyStrat category (one-sided burden P) |
|---|---|---|---|---|
| *LIPH*[a] | Lysophosphatidic acid production[61] | 7 | 6 (86%) | Overall ($4 \times 10^{-4}$) |
| *NRXN1* | Encodes a synapse adhesion molecule | 8 | 8 (100%) | Exon-Cons ($2 \times 10^{-4}$) |
| | | 13 | 10 (77%) | Exon-All ($2 \times 10^{-4}$) |
| | | 12 | 10 (83%) | Exon-Evo ($3 \times 10^{-4}$) |
| *HTR2A* | Indirect target for SSRI medications[50] | 4 | 4 (100%) | Exon-Cons ($9 \times 10^{-4}$) |
| *CTTNBP2* | Modulates postsynaptic cortactin[62] | 5 | 3 (60%) | DHS-Cons ($5 \times 10^{-4}$) |
| | | 15 | 8 (53%) | DHS-All ($9 \times 10^{-4}$) |
| *REEP3* | Regulates cellular vesicle trafficking | 3 | 3 (100%) | Overall-Div ($1 \times 10^{-4}$) |
| | | 4 | 4 (100%) | DHS-Evo ($6 \times 10^{-4}$) |
| | | 6 | 5 (83%) | DHS-All ($8 \times 10^{-4}$) |

[a]Significance of association possibly inflated by linkage between markers

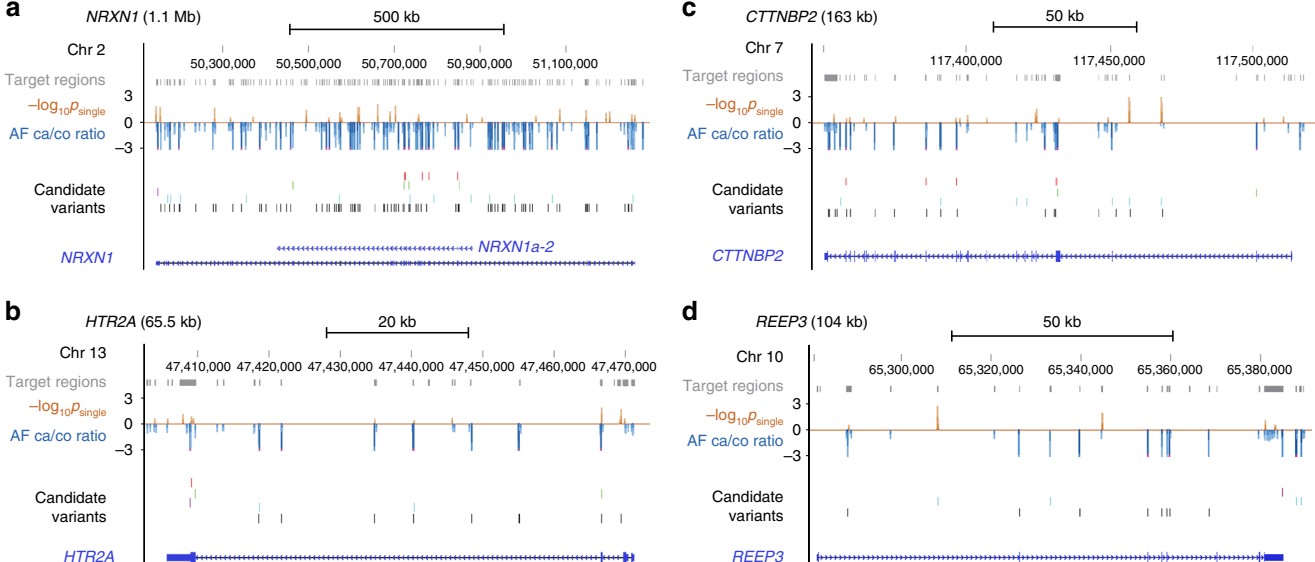

**Fig. 2** Targeted-sequencing detects both coding and regulatory candidate variants in the four top-scoring genes **a** NRXN1 **b** HTR2A **c** CTTNBP2 and **d** REEP3. Sequenced "Target regions" are shown as *gray boxes* above the *red* "$-\log_{10}p_{single}$" track displaying the association *p*-values for all detected variants and the *blue* "AF ca/co ratio" track showing the ratio of OCD AF over control AF. "Candidate variants" are annotated as missense (*red*), synonymous (*green*), untranslated (*purple*), DHS variants (*blue*), or unannotated (*black*). Lastly, the gene is shown as a *horizontal blue line* with exons (*solid boxes*) and *arrows* indicating direction of transcription. The highest scoring isoform of NRXN1, NRXN1a-2, is shown in (**a**)

PolyStrat considers 16 groups stratified by predicted function and evolutionary conservation.

PolyStrat *p*-values are corrected for multiple testing empirically using a permutation-based method that accurately measures experiment-wide statistical significance across correlated gene-based tests, while controlling for type 1 errors (Supplementary Methods). For most variant categories, quantile–quantile plots revealed good correspondence between observed values and the empirical null, with a small number of genes exceeding the expected distribution in a subset of the burden tests (Supplementary Figs. 3b and 4).

Five of the 608 sequenced genes (0.82%) show significant burdens of variants in OCD patients (Table 1; Fig. 1b), including two with excess coding variants (*NRXN1* and *HTR2A*) and two with excess regulatory variants (*CTTNBP2* and *REEP3*) (Fig. 2). *REEP3* is the only gene with excess divergent (potentially fast evolving) variants. No genes had a significant burden of rare variants (Supplementary Fig. 4).

We validated the 46 SNPs contributing to significant gene-burden tests (7 in *LIPH*, 13 in *NRXN1*, 4 in *HTR2A*, 15 in *CTTNBP2*, and 7 in *REEP3*) by individual genotyping of 571 OCD and 555 control samples (98% of the cohort). Nine variants failed Sequenom assay design or had low genotyping rates. For the remaining 37, the genotyping and pooled-sequencing frequencies are nearly perfectly correlated (Pearson's $\rho = 0.999$, $p < 2.2 \times 10^{-16}$; Supplementary Fig. 5; Supplementary Data 1).

We confirmed that our significant gene-burden test findings are not driven by population structure (Supplementary Methods) or linkage disequilibrium (LD), with one notable exception. We measured pairwise $r^2$ between SNPs contributing to the burden test in our top five genes, and found strong LD ($r^2 > 0.8$) between one pair, in *LIPH*. There was no strong LD in *NRXN1*, *HTR2A*, *CTTNBP2*, and *REEP3*.

Genes included from model-organism studies (263 genes) and larger ASD studies (196 genes) were significantly more associated than genes from human candidate gene studies (216 genes)

(Kruskal–Wallis $p = 5.6 \times 10^{-15}$). This is consistent with previous work showing that genes found through smaller candidate gene studies replicate poorly[11]. It also suggests that, when a genome-wide study of the disease of interest is not available, targeting genes implicated in a model organism may be as effective as targeting genes implicated in a comorbid, phenotypically similar human disorder.

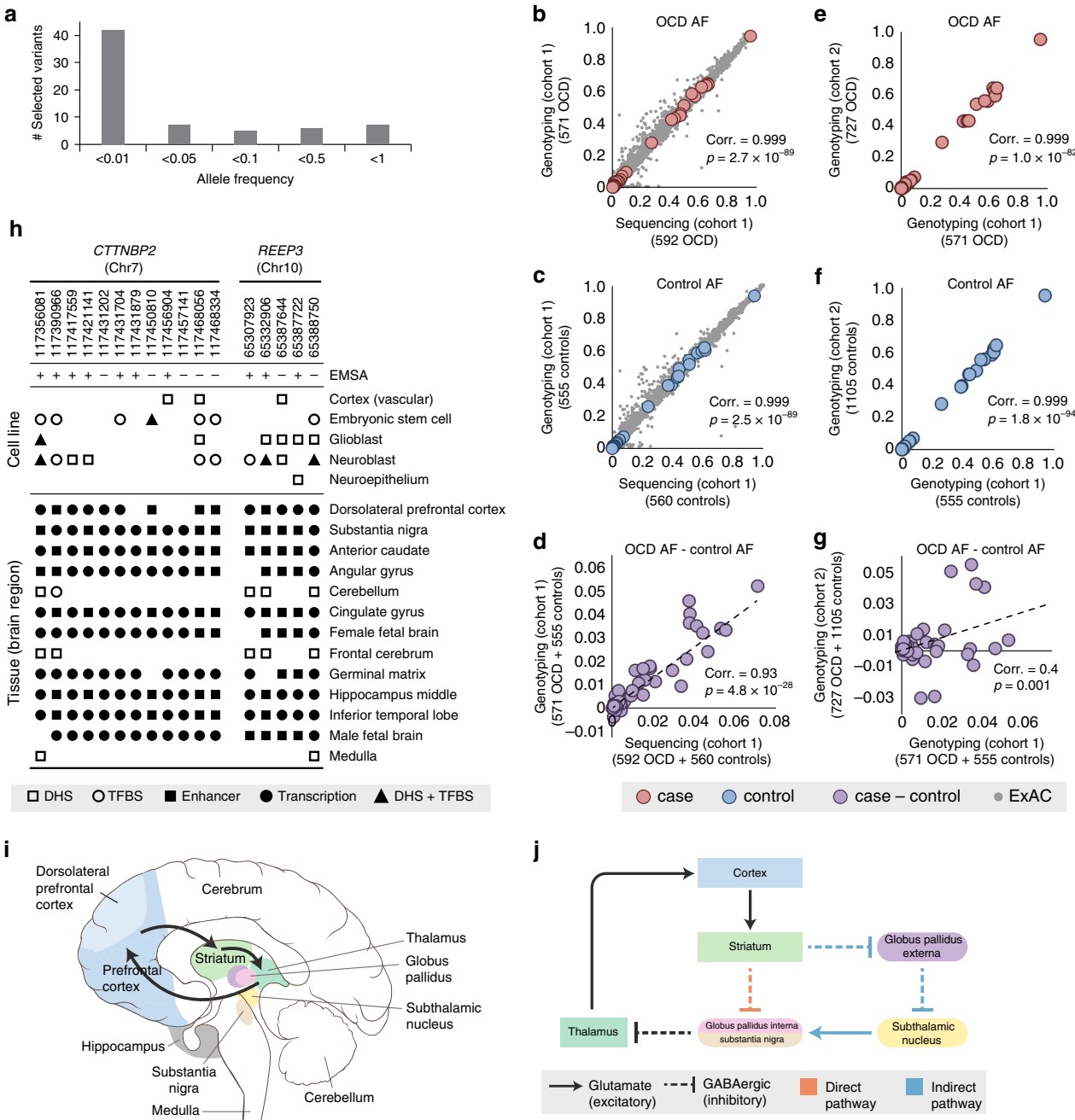

**Fig. 3** Validated top candidate variants disrupt functional elements active in brain. **a** Frequencies of the top candidate variants show that most are rare (AF < 0.01) in our cohort, illustrating the value of sequencing rather than array-based genotyping for detecting candidate variants. **b–g** Allele frequencies from pooled sequencing of individuals in the original cohort (cohort 1) are validated by genotyping in **b** cases and **c** controls, and by **d** allele-frequency differences between cases and controls. For the vast majority of variants, the pooled-sequencing allele frequencies are also highly correlated with frequencies observed in 33,370 ExAC individuals in both cases (**b**, gray dots) and controls (**c**, gray dots). Genotyping an independent cohort (cohort 2) of cases and controls reveals genotyping allele frequencies from cohort 1 are correlated in **e** cases and **f** controls; and for **g** allele-frequency differences between cases and controls. Correlation test was performed using Fisher's Z transform. **h** The top candidate variants in *CTTNBP2* and *REEP3*, the two genes enriched for regulatory variants, disrupt DNase hypersensitivity sites (DHS), enhancers, and transcription factor-binding sites (TFBS) annotated as functional in brain tissues and cell lines in ENCODE and Roadmap Epigenomics. All 17 variants disrupt elements active in either the dorsolateral prefrontal cortex and/or substantia nigra, which are among the **i** brain regions involved in the CSTC circuit implicated in OCD, illustrated with black arrows. Image adapted from Creative Commons original by Patrick J. Lynch and C. Carl Jaffe, MD. **j** The CSTC circuit requires a balance between a direct, GABAergic signaling pathway and an indirect pathway that involves both GABAergic and glutamate signaling. In OCD patients, an imbalance favoring the direct over the indirect pathway disrupts the normal functioning of the CSTC circuit[30]

The five genes most strongly implicated in canine CD and murine-compulsive grooming (*CDH2*, *CTNNA2*, *ATXN1*, *PGCP*, and *Sapap3*) have significantly lower *p*-values than the other 603 sequenced genes (Wilcoxon unpaired, one-sided $p = 2.6 \times 10^{-4}$). The difference becomes more significant when only rare variants are tested (Wilcoxon unpaired, one-sided $p = 3.2 \times 10^{-5}$) (Fig. 1c). This is consistent with the hypothesis that severe disease-causing variants, rare in humans due to negative selection, may persist at higher frequencies in model organisms where selection is relaxed.

Applying the burden test across multiple genes with shared biological functions, we identified gene sets with high-variant load in OCD patients. We tested all 989 Gene Ontology (GO) sets that are at least weakly enriched (enrichment $p < 0.1$) in our 608 sequenced genes (Supplementary Data 2) and found two with high-variant burdens: "GO:0010942 positive regulation of cell death" (uncorrected $p = 3 \times 10^{-4}$, corrected $p < 0.03$) and "GO:0031334 positive regulation of protein complex assembly" (uncorrected $p = 7 \times 10^{-4}$, corrected $p < 0.06$). Overlaying the burden test results onto the GO network topology highlights functional themes linking the enriched gene sets: regulation of protein complex assembly and cytoskeleton organization; neuronal migration; action potential; and cytoplasmic vesicle (Supplementary Fig. 6).

**Validation of candidate variants by genotyping**. We genotyped the top 67 candidate functional variants from the five significant genes, including 42 rare SNPs (AF < 0.01), in the pooled-sequencing cohort (Fig. 3a). This yielded, after QC, individual genotypes for 63 SNPs in 571 cases and 555 controls (98% of the cohort; genotyping rate >0.94 for all SNPs). We see near perfect correlation with the pooled sequencing for both allele frequencies (Fig. 3b, c; OCD AF, Pearson's $\rho = 0.999$, $p = 2.7 \times 10^{-89}$;

Control AF, Pearson's $\rho = 0.999$, $p = 2.5 \times 10^{-89}$) and the AF differences (Fig. 3d; OCD AF–control AF, Pearson's $\rho = 0.93$, $p = 4.8 \times 10^{-28}$).

We genotyped these 63 SNPs in an independent cohort of 727 cases and 1105 controls of European ancestry, and found strong correlation with the first genotyping cohort for both AF (Fig. 3e, f; OCD AF, Pearson's $\rho = 0.999$, $p = 1.0 \times 10^{-82}$; control AF, Pearson's $\rho = 0.999$, $p = 1.8 \times 10^{-94}$) and AF differences (Fig. 3g; OCD AF–control AF, Pearson's $\rho = 0.4$, $p = 0.001$). The risk allele from the first cohort is significantly more common in cases in the second cohort (Wilcoxon paired one-sided test for 63 SNPs, $p = 0.005$). More specifically, of 54 SNPs that had a higher frequency of the non-reference allele in cases in the first cohort, 61% also had a higher frequency of the non-reference allele in cases in the second cohort. The 33 SNPs that failed to validate in either of the two cohorts had smaller allele-frequency differences in the first cohort (one-sided unpaired *t*-test $p = 0.02$).

In summary, the allele-frequency analysis described above identified four genes: *NRXN1*, *HTR2A*, *CTTNBP2*, and *REEP3*. *LIPH* is excluded because its association is likely slightly inflated by LD and the genotyping in the second cohort did not reproduce as clearly. To validate the associations, we employed distinct strategies depending on whether the association was driven by coding (*NRXN1* and *HTR2A*) or regulatory variation (*CTTNBP2* and *REEP3*).

**Functional validation of regulatory variants using electrophoretic mobility shift assay**. For *CTTNBP2* and *REEP3*, regulatory variants give a far stronger burden signal than does testing for either coding variants or all variants (Fig. 1b). Furthermore, the three largest effect variants in the combined cohort (1298 OCD cases and 1660 controls) alter enhancer elements in these two genes:

**Table 2 Candidate regulatory variants**

| Chr:pos | Ref | Alt | rsID | Transcription factor | EMSA | OR | GERP |
|---|---|---|---|---|---|---|---|
| *CTTNBP2* | | | | | | | |
| chr7:117356081 | T | G | None | CTCF (GB, NB), RAD21 (NB, ESC) | c | Private[e] | b |
| chr7:117390966 | T | Del | None | CTCF (NB, CB), RAD21 (ESC) | d | Private[e] | b |
| chr7:117417559 | A | G | rs75322384 | b | c | 2.2 | Conserved |
| chr7:117421141 | C | A | None | b | d | Private[e] | Conserved |
| chr7:117431202[a] | C | A | None | b | b | Private[e] | Conserved |
| chr7:117431704[a] | C | T | None | RAD21 (ESC) | d | Private[e] | Conserved |
| chr7:117431879[a] | G | A | None | b | c | Private[e] | b |
| chr7:117450810 | C | T | rs34868515 | SP1, YY1, EP300, JUND, TCF12, HDAC2, NANOG, BCL11A, TEAD4 (all ESC) | b | Private | b |
| chr7:117456904 | C | T | rs12706157 | b | c | 1.06 | b |
| chr7:117457141 | G | C | rs13242822 | b | b | 1.04 | b |
| chr7:117468056 | C | T | rs2067080 | EP300 (NB), FOXP2 (NB), JUND (ESC) | b | 1.1 | Conserved |
| chr7:117468334 | T | C | rs2111209 | EP300 (NB), FOXP2 (NB), JUND (ESC) | b | 1.04 | b |
| *REEP3* | | | | | | | |
| chr10:65307923 | A | G | rs78109635 | GATA2 (NB) | c | 1.01 | Diverged |
| chr10:65332906 | T | C | rs76646063 | GATA3, GATA2, EP300 (all NB) | c | 3.7 | Conserved |
| chr10:65387644 | C | G | rs56311840 | b | b | Private[e] | b |
| chr10:65387722 | C | Del | None | b | d | Private[e] | b |
| chr10:65388750 | G | A | None | SIN3A (NB), POLR2A (NB), REST (NB), USF1 (ESC), EP300 (NB) | b | Private[e] | b |

We identified twelve candidate regulatory SNPs in *CTTNBP2*, including: seven intronic SNPs with DHS signals in neural stem cells (SK-N-MC) or neuroblasts (SK-N-SH, BE2-C, SH-SY5Y, SK-N-SH-RA), four of which also overlap TF-binding sites in the brain-derived cell lines; two intronic SNPs near the top DHS variants and potentially altering the same regulatory elements; and three coding SNPs that lie within or near regulatory marks (Supplementary Fig. 7b). We also identified five candidate regulatory SNPs in or near *REEP3*, including: one intronic SNP (chr10:65307923) in a DHS and GATA2 TFBS active in neuroblasts; one intronic SNP (chr10:65332906) that alters a DHS active in neural stem cells and GATA2, GATA3, and EP300 binding sites active in neuroblasts; three noncoding SNPs (chr10:65387644, chr10:65387722, and chr10:65388750) that cluster ~3 kb upstream in a DHS active in multiple brain-related cells, including neuroblasts, and are seen only in OCD patients in our pooled sequencing (Supplementary Fig. 7c).
[a]Coding; EMSA, electrophoretic mobility shift assay
[b]No change
[c]Strong TF-DNA binding change
[d]Weak change; GB, glioblast; NB, neuroblast; ESC, embryonic stem cell; CB, cerebellum; "Transcription factor" column shows the TF bindings to the regions and brain/developmental cell types where the signals are found. OR (odds ratio) column reports data in the combined set, unless noted with
[e]Indicating data from sequencing

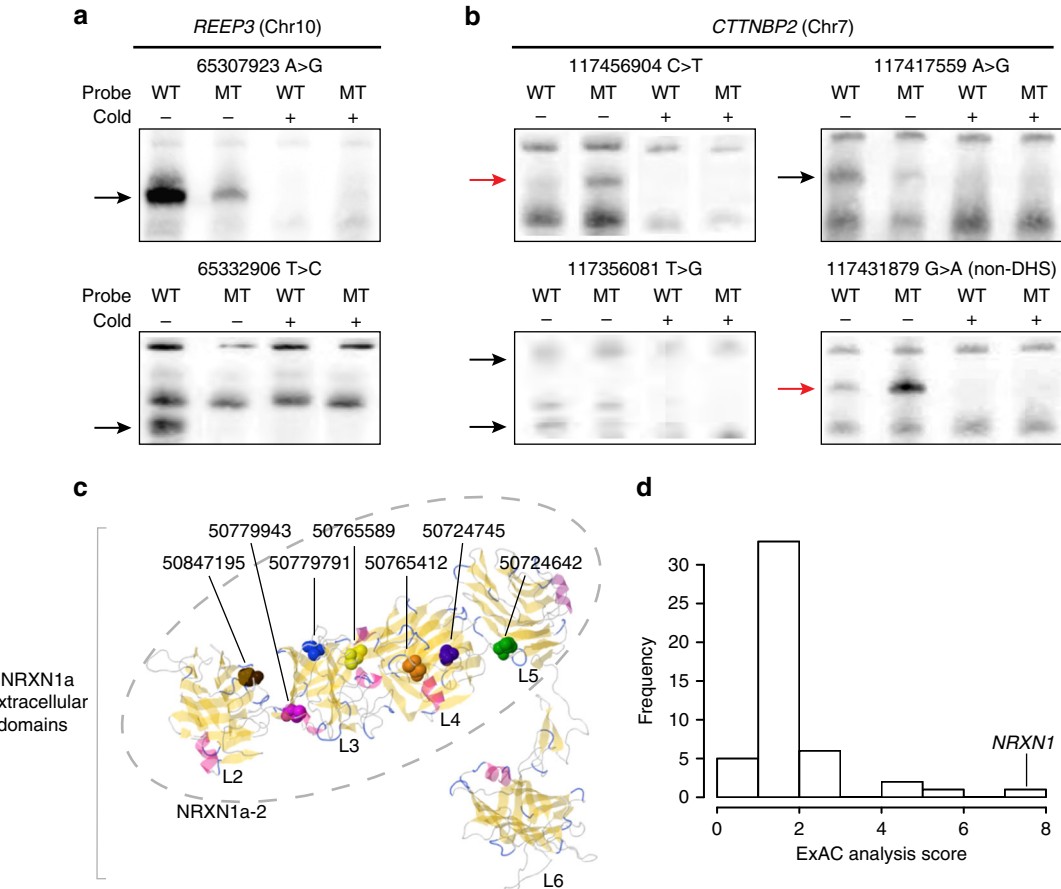

**Fig. 4** Top genes validate in functional assays in context of known protein structure and in comparison to ExAC. EMSA of all 17 candidate regulatory variants in **a** *REEP3* and **b** *CTTNBP2* reveals that six variants either decrease (*black arrows*) or increase (*red*) protein binding when the variant sequence (MT) is compared with the reference allele (WT). The signal disappears when competing unlabeled probes (Cold+) are added, confirming the specificity of the DNA-protein binding. Raw images as well as EMSA results for experimental replicates and for other candidate regulatory variants that showed weak-binding changes are shown in Supplementary Fig. 8. **c** All seven of the candidate missense variants in *NRXN1*, shown here as *colored* elements, alter the extracellular postsynaptic-binding region of our top-scoring protein isoform NRXN1a-2 [43, 59, 60]. Of the six extracellular LNS (laminin, nectin, sex-hormone-binding globulin) domains in the longer isoform NRXN1a, five with known protein structure are shown and labeled L2-L6. NRXN1a-2 includes four of these domains (L2–L5; *dashed ellipse*). **d** The isoform-based test comparing OCD cases to ExAC finds *NRXN1* as the top-scoring gene with genome-wide significance. The ExAC analysis score is defined as the ratio of $\chi^2$ statistics ($\sum_{i=1}^{n} \frac{(O_i - E_i)^2}{E_i}$, where $n$ = total number of isoforms, $O_i$ = number of non-reference alleles observed in isoform $i$, $E_i$ = number of non-reference alleles expected from ExAC in isoform $i$) between OCD vs. ExAC comparison and control vs. ExAC comparison

chr7:117358107 in *CTTNBP2* (OR = 5.2) and chr10:65332906 (OR = 3.7) and chr10:65287863 (OR = 3.2) in *REEP3* (Supplementary Data 3). Using ENCODE and Roadmap Epigenomics data, we identified 17 candidate SNPs in *CTTNBP2* and *REEP3*, likely to alter transcription factor-binding sites (TFBS) and/or disrupt chromatin structure in brain-related cell types [26, 29]. All 17 alter enhancers or transcription associated loci active in either the substantia nigra (SN), which relays signals from the striatum to the thalamus, and/or the dorsolateral prefrontal cortex (DL-PFC), which sends signals from the cortex to the striatum/thalamus (Fig. 3h, i). Both regions act in the CSTC pathway implicated by neurophysiological and genetic studies in OCD (Fig. 3j) [30].

We functionally tested 17 candidate regulatory SNPs in *REEP3* and *CTTNBP2* (Table 2; Supplementary Fig. 7b). We introduced each into a human neuroblastoma cell line (SK-N-BE(2)) and assessed transcription factor binding using electrophoretic mobility shift assays (EMSA). Both DHS SNPs in *REEP3*, three of seven DHS SNPs in *CTTNBP2*, and one non-DHS variant in *CTTNBP2* clearly alter specific DNA-protein binding (Fig. 4a, b). We see weak evidence of differential binding for one upstream

DHS SNP in *REEP3*, two DHS SNPs in *CTTNBP2*, and one non-DHS SNP in *CTTNBP2* (Supplementary Fig. 8).

The high rate of functional validation by EMSA demonstrates that screening using both regulatory and evolutionary information is remarkably effective in identifying strong candidate OCD-risk variants. In total, eight of 12 tested DHS SNPs (67%) show evidence of altered protein binding, despite testing a single cell line at a single time point under standard-binding conditions (Table 2). This includes two SNPs with high ORs in the full genotyping data sets that strongly disrupt specific DNA-protein binding (chr10:65332906 with OR = 3.7; chr7:117417559 with OR = 2.2). Two of five non-DHS SNPs (40%) also show altered binding, illustrating that DHS mark alone is a powerful but imperfect predictor of regulatory function. Both of these SNPs alter highly constrained elements (SiPhy score 8.7), whereas only one of the three non-DHS SNPs is constrained. Although this is a small data set, our results suggest that incorporating both DHS and conservation may identify functional regulatory variants with greater specificity, an observation consistent with previously published research [31].

**Table 3 Coding variants in *NRXN1* and *HTR2A***

| Chr:pos | Ref | Alt | rsID | OCD allele freq. | Ctrl allele freq. | ExAC allele freq. | Amino acid change | Candidate variant from sequencing |
|---|---|---|---|---|---|---|---|---|
| *NRXN1* | | | | | | | | |
| chr2:50149133 | C | T | rs113380721 | 0.0019 | 0.0030 | 0.0033 | Syn | No |
| chr2:50149214 | A | G | rs112536447 | 0.0001 | 0.0010 | 0.0003 | Syn | No |
| chr2:50280604 | T | C | rs79970751 | 0.0088 | 0.0070 | 0.0058 | Syn | No |
| chr2:50463984 | G | A | rs147580960 | 0.0009 | 0.0006 | 0 | Syn | Yes |
| chr2:50464065 | C | T | rs80094872 | 0.0009 | 0.0003 | 0[a] | Syn | Yes |
| chr2:50699479 | G | A | rs75275592 | 0.0009 | 0.0013 | 0.0010 | Syn | No |
| chr2:50723068 | G | A | rs56402642 | 0.0034 | 0.0001 | 0.0018 | Syn | Yes |
| chr2:50724642 | A | G | none | 0.0009 | 0 | 0 | I>T | Yes |
| chr2:50724745 | G | T | rs201818223 | 0.0017 | 0.0009 | 0.0011 | L>M | Yes |
| chr2:50733745 | G | C | rs147984237 | 0.0017 | 0 | $1.5 \times 10^{-5}$ | Syn | Yes |
| chr2:50765412 | G | T | rs56086732 | 0.0089 | 0.0028 | 0.0056 | L>I | Yes |
| chr2:50765589 | T | C | rs200074974 | 0.0016 | 0 | 0.0011 | I>V | Yes |
| chr2:50779791 | C | T | None | 0.0008 | 0 | 0[a] | A>T | Yes |
| chr2:50779943 | T | C | None | 0.0009 | 0 | 0[a] | N>S | Yes |
| chr2:50847195 | G | A | rs78540316 | 0.0077 | 0.0036 | 0.0043 | P>S | Yes |
| chr2:50850686 | G | A | rs2303298 | 0.0107 | 0.0013 | 0.0038 | Syn | Yes |
| *HTR2A* | | | | | | | | |
| chr13:47409048 | G | A | rs6308 | 0.0036 | 0.0012 | 0.0023 | A>V | Yes |
| chr13:47409701 | G | A | rs141413930 | 0.0020 | 0.0012 | 0.0026[a] | Syn | Yes |
| chr13:47409149 | T | A | rs35224115 | 0.0019 | 0.0047 | 0.0044 | Syn | No |
| chr13:47466622 | G | A | rs6305 | 0.0386 | 0.0225 | 0.027[b] | Syn | Yes |

ExAC allele freq. shown for NFE population
[a]Excluded from ExAC analysis because of low-confidence call from pooled sequencing
[b]Excluded from ExAC analysis because of frequency >0.01

**Validation of coding variants using ExAC**. In contrast to the regulatory-variant burden found in *CTTNBP2* and *REEP3*, *NRXN1* and *HTR2A* showed significant PolyStrat signals when only coding variants are considered. Of 12 candidate coding SNPs in *NRXN1*, seven are missense (Table 3). Four of these are SNPs private to OCD cases, and the other three are rare (AF in controls 0.0009–0.0036). All seven change amino acids in laminin G or EGF-like domains important for postsynaptic binding, potentially affecting the involvement of NRXN1 in synapse formation and maintenance (Fig. 4c). Of the three candidate coding SNPs in *HTR2A*, two (one missense and one synonymous) are in the last coding exon, and one (missense) is the cytoplasmic domain with a PDZ-binding motif, potentially affecting binding affinity or specificity[32].

We sought to improve our statistical power by combining our pooled-sequencing data with publicly available ExAC data[33]. Using only our data, the associations of *CTTNBP2*, *REEP3*, *NRXN1*, and *HTR2A* with OCD are experiment-wide significant, but do not reach the genome-wide significance threshold $p < 2.5 \times 10^{-6}$ (~20,000 human genes), with the strongest association, to *NRXN1*, at $p = 5.1 \times 10^{-5}$ (cohort 1 and 2; Fisher's combined $p$). For the two genes with a burden of coding variants (*NRXN1* and *HTR2A*), we used ExAC to assess variant burden in OCD cases compared with 33,370 non-Finnish Europeans. Such a comparison was not possible for *CTTNBP2* and *REEP3*, for which associated variants are predominantly noncoding and thus not assayed in ExAC.

To assess the significance of the variant enrichment in each gene, we used an isoform-based test that incorporates a within-gene comparison to assess significance, effectively controlling for inflation due to the extremely large size of the ExAC cohort[34] (Supplementary Methods). Of 542 genes with more than one isoform, we saw no significant difference between our control data and ExAC for over 90% (493 genes had corrected $p > 0.05$). Focusing on the subset of 66 genes with nominally significant PolyStrat scores, *NRXN1* had the largest difference between cases

and ExAC ($\chi^2 = 82.3$, df = 16, uncorrected $p = 6.37 \times 10^{-11}$; corrected $p = 1.27 \times 10^{-6}$) and no difference between controls and ExAC ($\chi^2 = 10.5$, df = 16, uncorrected $p = 0.84$) (Fig. 4d). No previous findings in OCD genetics have reached this level of significance despite >100 candidate gene studies[35], a dozen linkage studies[30], and two GWAS[2, 3]. *HTR2A*, while enriched for coding variants, had only two SNPs in cases, providing insufficient information for the isoform test.

The significant association of *NRXN1* reflects an exceptional burden of variants in one of its 17 Ensembl isoforms. *NRXN1a-2*, which contains all 12 candidate coding SNPs, had the largest deviation between observed and expected variant counts, with a residual at least 1.4× higher than any other isoform (*NRXN1a-2* = 22.3, *NRXN1-001* = 16.3; median = 5.15). After adjusting for the residuals from the "null" control data and ExAC comparison, the *NRXN1a-2* residual is still 1.3× higher (OCD residual/control residual *NRXN1a-2* = 5.34, *NRXN1-014* = 4.04).

## Discussion

By analyzing sequencing data for 608 OCD candidate genes, then prioritizing variants according to functional and conservation annotations, we identified four genes with a reproducible variant burden in OCD cases. Two genes, *NRXN1* and *HTR2A* (Table 3), have a burden of coding variants, and the other two, *CTTNBP2* and *REEP3* (Table 2), have a burden of conserved regulatory variants. Notably, all four act in neural pathways linked to OCD, including serotonin and glutamate signaling, synaptic connectivity, and the CSTC circuit[6], offering new insight into the biological basis of compulsive behavior (Fig. 5).

We used three independent approaches to validate our findings: (1) For the top candidate SNPs, allele-frequency differences from sequencing data were confirmed by genotyping of both the original cohort (Fig. 3d) and a larger, independent cohort (Fig. 3g). (2) For the two genes with a burden of coding variants (*NRXN1* and *HTR2A*), comparison of our data to 33,370 population-matched controls from ExAC[33] revealed genome-

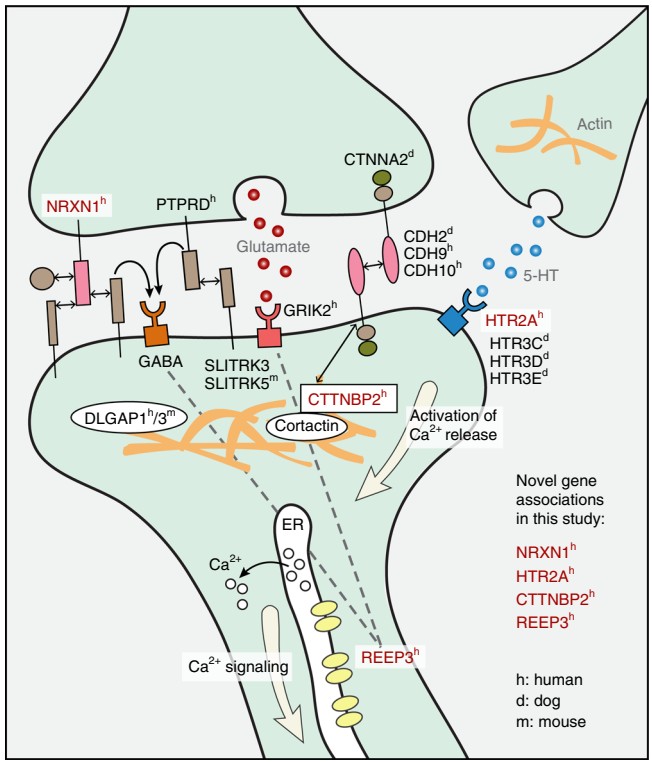

**Fig. 5** All four top candidate genes function at the synapse and interact with proteins implicated in OCD by previous studies in the three species. Human, dog, and mouse are marked with *superscripts h, d*, and *m*, respectively. Genes identified in this study are shown in *red*. *Solid lines* indicate direct interactions, and *dashed lines* indicate indirect interactions

wide-significant association of *NRXN1* with OCD. (3) For the two genes with the burden of regulatory variants (*REEP3* and *CTTNBP2*), more than one-third of candidate SNPs altered protein/DNA binding in a neuroblastoma cell line (Fig. 4a).

Comparison of our approach to existing methods illustrates its unique advantages, and offers a deeper understanding of how its two key features—targeted sequencing, and incorporation of functional and conservation metrics—permit identification of significantly associated genes using a cohort smaller even than those that have previously failed to yield significant results.

Targeted sequencing captures both coding and regulatory variants, and both common and rare variants, at a fraction of the cost of whole-genome sequencing (WGS). For the modest-size cohort in this study, WGS would cost ~$2.5M, 40-fold more than our pooled-sequencing approach. Even without pooling, our targeted-sequencing costs fourfold less than WGS. Whole-exome sequencing would cost approximately the same as targeted sequencing, but misses the regulatory variants explaining most polygenic trait heritability[21]. By using existing information on OCD and related diseases to prioritize a large set of genes, then performing targeted sequencing of functional elements, our approach enhances causal-variant detection and thus statistical power, although it misses OCD-associated genes not included as candidates, and potential distant regulatory elements.

The capacity to detect associations to rare variants is especially critical for study of diseases that, like OCD, may reduce fitness, as negative selection limits inheritance of deleterious variants[36]. Genotype array data sets, and even imputed data sets, miss many rare variants. In our data set, 80% of variants driving significant associations have allele frequencies <0.05; one of the densest genotyping arrays available, the Illumina Infinium Omni5 (4.3M markers) contains only half of these variants (Supplementary

Data 1)[2, 3]. In addition, 60% of our variants have allele frequencies <0.01, and would be missed even through imputation with 1000 Genomes and UK10K[37].

Our new analytical method, Poly Strat, analyzes targeted-sequencing data capturing all variants, and leverages public evolutionary and regulatory data to increase power. PolyStrat first filters out variants that are less likely to be functional, then performs gene-burden tests. In contrast to gene-based approaches focusing on ultra-rare, protein-damaging variants, PolyStrat considers variants of diverse frequencies, gaining power to identify genes with excess variants in cases.

PolyStrat is particularly advantageous when applied to studies with smaller cohorts. By testing for association at the gene level, it requires statistical correction only for the ~20,000 genes in the genome. It increases power further by using targeted-sequencing data to capture nearly all variation, including variants with higher allele frequencies and/or larger effect sizes, in regions that are coding or evolutionarily constrained, and enriching for causal variants by removing ~33% of variants unlikely to be functional[38]. PolyStrat tests ~82 times more functional variants than PolyPhen2 (http://genetics.bwh.harvard.edu/pph2/), which focuses on protein-damaging variants (27,626 vs. 335 in our data).

Our PolyStrat results are consistent with expectations from simulations, which suggest that 200–700 cases should yield 90% power to detect associated genes with allele frequencies and effect sizes similar to our four genes[39]. Specifically, we would achieve 90% power to detect associations to *NRXN1* (combined AF = 0.022, OR = 2.4) with ~600 cases, to *HTR2A* (combined AF = 0.03, OR = 1.56) with ~700 cases; to *REEP3* (combined AF = 0.04, OR = 2.11) with ~200 cases, and to rare (AF < 0.01) variants in *CTTNBP2* (combined AF = 0.003, OR = 4.7) with ~500 cases.

Previous research on the four genes identified by PolyStrat revealed that all are expressed in the striatum, a brain region linked to OCD (http://human.brain-map.org/). All four genes are involved in pathways relevant to brain function, and harbor variants that could alter OCD risk (Table 4).

*NRXN1* encodes the synapse cell-adhesion protein neurexin 1α, a component of cortico-striatal neural pathway[40, 41] implicated in ASD and other psychiatric diseases[42], and functionally related to genes associated with OCD (*CDH9/CDH10*)[3, 8, 9] and canine CD[8, 9] (*CDH2*) (Fig. 5). *NRXN1* isoforms are implicated in distinct neuropsychiatric disorders. The non-synonymous variants in the *NRXN1a-2* isoform (Fig. 4c) may alter synaptic function by disrupting cellular localization or interactions with binding partners, including neurexophilins[43]. The five synonymous candidate variants in likely regulatory elements may affect protein folding by disrupting post-transcriptional regulation, seen in other neuropsychiatic disorders[44].

The synaptic plasticity gene *REEP3*, also implicated in ASD[45], encodes a protein that shapes tubular endoplasmic reticulum membranes found in highly polarized cells, including neurons[46]. The two EMSA-validated *REEP3* variants change regulatory elements active in the cortico-striatal neural pathway (Fig. 3h) and bound by multiple TFs (Table 2) including GATA2, which may be required to actuate inhibitory GABAergic neurons[47]. Thus, variants disrupting GATA2 binding could change the balance between excitatory and inhibitory neurons in the CSTC circuit (Fig. 3j)[30].

*CTTNBP2* regulates postsynaptic excitatory synapse formation. All four EMSA-confirmed variants in *CTTNBP2* alter epigenetic marks active in the key structures of the cortico-striatal neural pathway[48] (Table 2; Fig. 3h), potentially affecting the expression of this critical gene. CTTNBP2 proteins interact with both proteins encoded by *STRN* (striatin), which approached experiment-wide significance in this study (uncorrected *p* = 0.0016, corrected *p* < 0.1; Fig. 1b) and the canine CD gene *CDH2* (Fig. 5).

**Table 4 Summary of top genes**

|  | NRXN1 | REEP3 | CTTNBP2 | HTR2A |
|---|---|---|---|---|
| Gene product/ brain relevance | Synaptic cell-adhesion protein/ synapse functioning and development in the cortical-striatal pathway[40, 41] | Microtubule-binding protein/ possible role in synaptic plasticity, calcium signaling, shaping tubular ER membranes in neurons[46] | Cortical actin (cortactin)-binding protein/synaptic maintenance[48] | G-protein-coupled serotonin receptor/cortical neuron excitation[63] |
| Disease relevance | Neurodevelopmental disorders incl. ASD[42, 64] | ASD[45] | Interacts with CDH2, implicated in canine CD[8, 9, 65] | ASD, OCD[35], canine CD (5-HT3 receptors)[49] |
| Reason for inclusion as candidate[a] | (1) Model (mouse) organism gene; (2) ASD gene | (2) ASD gene | (1) Model (mouse) organism gene | (1) Model (dog) organism gene; (2) human candidate—SSRI target |
| Type of burden identified | Coding variants—missense variants over-represented to one isoform, NRXN1a-2 | Regulatory variants | Regulatory variants | Coding variants—missense variant in perfect linkage to a common variant rs6314 associated with response to SSRIs[50] |
| Validation of variants identified in present study Hypothesized impact of variants identified | By comparison to ExAC—genome-wide-significant association Inaccurate cellular localization of NRXN1, or altered binding competition to its partner, modifying synaptic adhesion[43] | By EMSA—disrupt regulatory elements bound by various TF, including GATA2 REEP3 expression in GABA neurons inhibited by variants that reduce GATA2 binding[47], leading to excitatory/inhibitory imbalance in CSTC circuit[30] | By EMSA—alter epigenetic marks active in the cortico-striatal pathway Altered CTTNBP2 expression in cortical-striatal circuit in brain[66] | By genotyping independent cohort; too few polymorphic sites for validation with ExAC Altered binding affinity of HTR2A, changing the activation of downstream calcium signaling in neurons[32] |

[a]For explanation of each category, see "Results" section

HTR2A encodes a G-protein-coupled serotonin receptor expressed throughout the central nervous system, including in the prefrontal cortex, and has been implicated in ASD and OCD[35]. A related serotonin-receptor cluster (HTR3C/HTR3D/HTR3E) is associated with severe canine CD[49] (Fig. 5). The three coding variants found in HTR2A may alter its binding affinity (Table 3)[32], and one of the three, a rare missense variant (rs6308; AF = 0.004 in 1000G CEU population) is perfectly linked ($D' = 1$; http://raggr.usc.edu) to a common variant (rs6314) associated with response to SSRIs[50].

Taken together, our top four associated genes and our pathway analysis implicate three classes of neuronal functions in OCD, as described below.

First, synaptic cell-adhesion molecules help establish and maintain contact between the presynaptic and postsynaptic membrane, and are critical for synapse development and neural plasticity. NRXN1 encodes a cell-adhesion molecule predominantly expressed in the brain, and CTTNBP2 regulates cortactin, another such molecule, echoing earlier findings linking cell-adhesion genes to compulsive disorders in dogs (CDH2 and CTNNA2), mice (Slitrk5), and humans (DLGAP1, PTPRD and CDH9/CDH10)[2, 3, 8, 51] (Fig. 5). In our pathway analysis, "regulation of protein complex assembly" and "cytoskeleton organization" were enriched for variants in OCD patients.

Second, OCD may result from an imbalance of excitatory glutamate and inhibitory GABAergic neuron differentiation[30] (Fig. 3j), a process that involves both NRXN1[52] and REEP3[53] (Table 4), as well as PTPRD, a top OCD GWAS candidate[3]. We also find an overall burden of variants in genes regulating cell death and apoptosis (Supplementary Data 2) and in telencephalic tangential migration, a neuronal migration event which forms connections between the key structures of CSTC circuit[54].

Third, SSRIs are the most effective available OCD treatment, suggesting a role for serotonergic pathways in disease. HTR2A encodes a serotonin receptor, and allelic variation in HTR2A alters response to SSRIs (Table 4)[50]. In addition, both REEP3 and CACNA1C, which score high in this study (Fig. 1), also

significantly associate with schizophrenia and act in calcium signaling, a downstream pathway of HTR2A[55–57]. Meta-analysis of >100 OCD genetic association studies found strong association to both HTR2A and the serotonin transporter gene SLC6A4[35]. In dogs, a serotonin-receptor locus is associated with severe CD[49].

Our findings suggest broad principles that could guide studies of other polygenic diseases. We discovered that genes associated in selectively bred model organisms are more likely to contain rare, highly penetrant variants. The five genes we found to be most strongly associated with compulsive behaviors in dog and mouse (CDH2, CTNNA2, ATXN1, PGCP, and Sapap3) were significantly more enriched for rare variants in human patients than the other 603 genes targeted, although they did not individually achieve significance (Fig. 1c). We propose that the enrichment of rare variants in humans reflects natural selective forces limiting the prevalence of severe disease-causing variants. Such forces are less powerful in selectively bred animal populations. Because risk variants identified through animal models are anticipated to be rare in humans, replication will require either family-based studies, or cohorts of magnitude not currently available.

We also find that the ratio of coding to regulatory variants is positively correlated with a gene's developmental importance. Although single-gene $p$-values from PolyStrat tests are positively correlated across variant categories, as is expected given overlaps between different variant categories (Fig. 1a; Supplementary Fig. 9), this pattern breaks down for our four significantly associated genes. NRXN1 and HTR2A, which have burdens of coding variants, score poorly on regulatory-variant tests; CTTNBP2 and REEP3, which have burdens of regulatory variants, score poorly in coding-variant tests (Fig. 1b). This is consistent with the ExAC study showing that genes critical to viability or development do not tolerate major coding changes[33]. In that study, the authors infer that CTTNBP2 and REEP3 would be intolerant of homozygous loss of function variants ($p$Rec = 0.99999015 and $p$Rec = 0.953842585, respectively), whereas HTR2A ($p$Rec = 0.225555783) and, most notably, NRXN1 ($p$Rec

$= 5.13 \times 10^{-5}$) would be far more tolerant. Our finding of enrichment for regulatory variants in *CTTNBP2* and *REEP3* suggests that these genes may tolerate variants with more subtle functional impacts, such as expression differences in specific cell types or developmental stages.

Technological advances in high-throughput sequencing bring increased focus on identifying causal genetic variants as a first step toward targeted disease therapies[58]. However, existing approaches have notable limitations. WGS is prohibitively expensive in large cohorts, whereas cost-saving whole-exome sequencing does not capture the regulatory variants underlying complex diseases[21]. Leveraging existing genomic resources can increase power to find causal variants through meta-analysis and imputation, but these resources are heavily biased towards a few populations. Without new approaches, advances in precision medicine will predominantly benefit those of European descent.

Here, we describe an approach that combines prior findings, targeted sequencing, and a new analytic method to efficiently identify genes and individual variants associated with complex disease risk. In a modest-size cohort of OCD cases and controls we find associations driven by both coding and regulatory variants, highlighting new potential therapeutic targets. Our method holds promise for elucidating the biological basis of complex disease, and for extending the power of precision medicine to previously excluded populations.

## Methods

**Study design**. We designed and carried out the study in two phases. In the first, discovery phase, we performed targeted sequencing of 592 individuals with DSM-IV OCD and 560 controls of European ancestry, and tested association for OCD at single variant-level, gene-level, and pathway-level. In the second, validation phase, we employed three distinct analyses. (1) We genotyped both the original cohort and a second, independent cohort containing 1834 DNA samples (729 DSM-IV OCD cases and 1105 controls) of European ancestry, including a total of 2986 individuals (1321 OCD cases and 1665 controls) to confirm the observed allele frequencies in the discovery phase. (2) We compared our sequencing data with 33,370 population-matched controls from ExAC to confirm the gene-based burden of coding variants as well as allele frequencies. (3) We performed EMSA to test whether our candidate variants have regulatory function. Uses of biospecimens in this study were reviewed and approved by either the Broad's Office of Research Subject Protection, or the Partners HealthCare Human Research Committee. Informed consent was obtained from all subjects included in our study.

**Targeted regions**. We targeted 82,723 evolutionarily constrained regions in and around 608 genes, which included all regions within 1 kb of the start and end of each of 608 targeted genes with SiPhy evolutionarily constraint score >7, as well as all exons[22]. For the intergenic regions upstream and downstream of each gene, we used constraint score thresholds that became more stringent with distance from the gene.

**Pooled sequencing and variant annotation**. Groups of 16 individuals were pooled together into 37 case pools and 35 control pools and bar-coded. Targeted-genomic regions were captured using a custom NimbleGen hybrid capture array and sequenced by Illumina GAII or Illumina HiSeq2000. Sequencing reads were aligned and processed by Picard analysis pipeline (http://broadinstitute.github.io/picard/). Variants and AFs were called using Syzygy[17] and SNVer[23]. We used ANNOVAR[25] to annotate variants for RefSeq genes (hg19), GERP scores, ENCODE DHS cluster, and 1000 G data.

**Genotyping**. SNP genotyping was performed using the Sequenom MassARRAY iPLEX platform. The resulting data were analyzed using PLINK1.9 (www.cog-genomics.org/plink2).

**EMSA**. For each allele of the tested variants, pairs of 5′-biotinylated oligonucleotides were obtained from IDT Inc. (Supplementary Data 4). Equal volumes of forward and reverse oligonucleotides (1 pmol/μl) were mixed and heated at 95°C for 5 min and then cooled to room temperature. Annealed probes were incubated at room temperature for 30 min with SK-N-BE(2) nuclear extract (Active Motif). The remaining steps followed the LightShift Chemiluminescent EMSA Kit protocol (Thermo Scientific).

**Statistical analysis**. For gene-association/pathway-association, we used the sum of the differences of non-reference allele rates between cases and controls per gene as test statistic, and calculated the probability of observing a test statistic by chance from 10,000 permutations. Multiple testing was empirically corrected using "minP" procedure. See Supplementary Methods for details.

**Code availability**. The code used in this study was obtained from R package Rplinkseq and PLINK1.9.

**Data availability**. All data presented in this study are accessible at: https://data.broadinstitute.org/OCD_NatureCommunications2017/.

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

## Acknowledgements

We thank the participating individuals for their support, Eric S. Lander, Steven E. Hyman, Jessica Alföldi, and Kaitlin Samocha for valuable input; Leslie Gaffney for help with illustrations; Jeremiah M. Scharf for sample contribution and discussions; and Broad Genomics Platform for sample processing, sequencing, and genotyping. H.J.N. is supported by the AKC Health Foundation and Swedish Research Council, C.R. by the Swedish Research Council (K2013-61P-22168), K.L.-T. by the Swedish Medical Research Council and European Research Council, and E.K.K. by NIH NIMH (1R21MH109938-01). A Broad Institute SPARC grant supported part of this work.

## Author contributions

K.L.-T., E.K.K., G.F., H.J.N., and R.T. conceived and designed the experiments. H.J.N., R.T., E.K.K., J.F., C.O.'D., R.S., D.H., D.P.G., K.L.-T. analyzed the data. H.J.N., D.P.G., E.K.K., and K.L.-T. wrote the paper. R.T. and H.J.N. performed sequence capture. R.S. performed EMSA. M.W., H.-J.G., S.R., C.A.M., S.E.S., S.A.R., M.A.J., J.A.K., C.R., E.G., G.L.H., D.C.C., E.A., S.W., P.D.P., C.H., M.T.P., and C.N.P. diagnosed/collected samples. J.J., M.K., and G.v.G. coordinated/prepared samples and data generation.

## Additional information

**Competing interests:** The authors declare is no competing financial interests.

Hyun Ji Noh[1], Ruqi Tang[1,2,3], Jason Flannick[1], Colm O'Dushlaine[1], Ross Swofford[1], Daniel Howrigan[1], Diane P. Genereux[1], Jeremy Johnson[1], Gerard van Grootheest[4], Edna Grünblatt[5,6,7], Erik Andersson[8], Diana R. Djurfeldt[8,9], Paresh D. Patel[10], Michele Koltookian[1], Christina M. Hultman[11], Michele T. Pato[12], Carlos N. Pato[12], Steven A. Rasmussen[13], Michael A. Jenike[14], Gregory L. Hanna[10], S.Evelyn Stewart[15], James A. Knowles[12], Stephan Ruhrmann[16], Hans-Jörgen Grabe[17], Michael Wagner[18,19], Christian Rück[8,9], Carol A. Mathews[20], Susanne Walitza[5,6,7], Daniëlle C. Cath[21], Guoping Feng[2,22], Elinor K. Karlsson[1,23] & Kerstin Lindblad-Toh[1,24]

[1]Broad Institute of MIT and Harvard, 415 Main Street, Cambridge, MA 02142, USA. [2]Department of Brain and Cognitive Sciences, McGovern Institute for Brain Research, Massachusetts Institute of Technology, 43 Vassar Street, Cambridge, MA 02139, USA. [3]Renji Hospital, School of Medicine, Shanghai Jiao Tong University, 145 Shandong Middle Road, Huangpu Qu, Shanghai 200001, China. [4]GGZ inGeest and Department of Psychiatry, VU University Medical Center, De Boelelaan 1117, 1081 HV, Amsterdam, The Netherlands. [5]Department of Child & Adolescent Psychiatry and Psychotherapy, University Hospital of Psychiatry Zurich, University of Zurich, Neumünsterallee 9, Zurich 8032, Switzerland. [6]Neuroscience Center Zurich, University of Zurich & ETH Zurich, Winterthurer Strasse 190, Zurich 8057, Switzerland. [7]Zurich Center for Integrative Human Physiology, University of Zurich, Winterthurer Strasse 190, Zurich 8057, Switzerland. [8]Department of Clinical Neuroscience, Centre for Psychiatry Research, Karolinska Institutet Tomtebodavägen 18A, Stockholm 17177, Sweden. [9]Stockholm Health Care Services, Stockholm County Council, Stockholm 14186, Sweden. [10]Department of Psychiatry, University of Michigan, 4250 Plymouth Road, Ann Arbor, MI 48109, USA. [11]Department of Medical Epidemiology & Biostatistics, Karolinska Institutet, Stockholm 17177, Sweden. [12]Department of Psychiatry & Behavioral Sciences, USC, 2250 Alcazar Street, Los Angeles, CA 90033, USA. [13]Department of Psychiatry & Human Behavior, Brown Medical School, 345 Blackstone Boulevard, Box G-BH, Providence, RI 02906, USA. [14]Department of Psychiatry, Harvard Medical School, 401 Park Drive, Boston, MA 02215, USA. [15]BC Mental Health & Addictions Research Institute, UBC, 2255 Wesbrook Mall, Vancouver, BC, Canada V6T 2A1. [16]Department of Psychiatry & Psychotherapy, University of Cologne, Kerpener Street 62, Cologne 50937, Germany. [17]Department of Psychiatry & Psychotherapy, University of Medicine Greifswald, Fleischmannstrasse 8, Greifswald 17475, Germany. [18]Department of Psychiatry & Psychotherapy, University of Bonn, Regina-Pacis-Weg 3, Bonn 53113, Germany. [19]German Center for Neurodegenerative Diseases, Sigmund-Freud-Strasse 27, Bonn 53127, Germany. [20]Department of Psychiatry & Genetics Institute, University of Florida, 1149 Newell Drive, Gainesville, FL 32610, USA. [21]Department of Clinical & Health Psychology, Utrecht University, Heidelberglaan 1, Utrecht, CS 3584, The Netherlands. [22]Stanley Center for Psychiatric Research, Broad Institute of MIT and Harvard, 415 Main Street, Cambridge, MA 02142, USA. [23]Program in Bioinformatics & Integrative Biology and Program in Molecular Medicine, University of Massachusetts Medical School, 368 Plantation Street, Sherman Center, Worcester, MA 01605, USA. [24]Science for Life Laboratory, IMBIM, Uppsala University, Uppsala 75236, Sweden. Elinor K. Karlsson and Kerstin Lindblad-Toh contributed equally to this work

