## [Peer Review File · Nature Communications]

Reviewers' comments:

Reviewer #1 (Remarks to the Author):

In this report, Noh and colleagues successfully employ a novel approach to identify the first genome-wide significant finding for OCD (i.e. a rare coding variant burden in NRXN1, $p=6.37 \times 10^{-11}$). Historically, this is a very significant accomplishment, because no other findings in OCD genetics have reached this level of significance despite >100 candidate gene studies (meta-analyzed by Taylor et al, PMID 22665263), a dozen linkage studies (reviewed by Pauls et al, PMID 24840803) and two genome-wide association studies (PMIDs 22889921 and 24821223). The authors may wish to emphasize this point. The authors also introduce an innovative method that both integrates cross-species (mouse/dog/human) genetic data and provides a cost- and analytically-efficient approach to examine conserved non-coding elements. It is impressive to see this work on a behavioral trait and it will be very interesting to see if this method extends to other phenotypes (behavioral and otherwise).

Strengths:

1. Despite a somewhat complicated, multi-step approach, the paper remains easy to follow and the results are clear.
2. Rigor - the authors performed careful validation of pooled genotype findings (which can often cause problems) and included replication within this study as well as relating coding variants to the broader literature through ExAC.
3. Despite using a candidate gene approach and targeted resequencing, the authors provided a strong justification for the genes (and associated gene regions) included.
4. Overall results from Figure 1c, demonstrating that this candidate gene set is enriched for meaningful associations.
5. Functional verification of regulatory variants via EMSA.

Weaknesses:

1. Many of the coding variants associated with OCD are synonymous. The authors should comment on how they envision these impacting gene function.

Reviewer #2 (Remarks to the Author):

This study used targeted sequencing and a new analysis method to identify genes and individual variants associated with OCD risk, and validate the result in independent data and by experiments, which may extend the power to elucidate the biological basis of OCD.

My concerns are:

1. How to select candidate genes for targeted sequencing is very important in this paper, but it's not clear from author's description: 1) "56 genes from regions associated in our GWAS of canine CD", how to define the regions associated in GWAS, the author gave two citations but how big the region is and why chose that length should be provided; 2) for ASD genes, I don't know which "whole genome study of ASD in humans" the author used (no citation was provided), the sample size and statistical power

are big issues to consider; 3) for the human candidate genes, why author selected SCZ genes found in 2008 and 2009, but not included recent progress by PGC in past three years?

2. Gene length is a confounding factor when counting the number of SNPs, longer genes tend to carry more SNPs theoretically. But no clear adjustment was provided in this study, which will produce bias result. Same problem exists when we compare the regulation regions of those genes.

3. The variant categories in PolyStrat were really confusing: what is the difference between "(i) all" and "(iv) 'evolutionary' (Evo) combining (ii) and (iii)" in evolution categories? The criteria of Cons and Div should be provided.

4. In the EMSA validation section, if author proposed "incorporating both DHS and conservation" can better predict functional regulatory variants, they should construct a statistical model and provide more convincing data and analysis, such as the sensitive and specificity results.

5. Striatum is an important region linked to OCD, which author mentioned in summary part, but whether the strongly associated genes they found active in this region is not provided.

6. The ExAC database is now updated to gnomAD (<http://gnomad.broadinstitute.org>), with more than double sample sizes and data. Comprehensive coding variants' validation should use gnomAD database, especially for rare variants.

7. Author claimed this method can "highlighting new potential new therapeutic targets" in introduction, but to be precise, the targets are not new when using capture sequencing. This method can validate existing candidate genes, and explore new variants in coding and regulatory regions.

REVIEWERS' COMMENTS:

Reviewer #1 (Remarks to the Author):

The authors have addressed my concerns and I view the current version as acceptable for publication.

Reviewer #2 (Remarks to the Author):

The authors have addressed all my comments, and the answers are acceptable.

Point by point response to comments from the two reviewers.

We would like to thank the reviewers for taking the time to review our manuscript and to provide detailed comments. We are glad that both reviewers acknowledged the power of our novel approach in elucidating the genetic and biological basis of OCD. In order to address the reviewers' comments, we have revised the text and conducted additional analyses, as explained below.

Reviewer #1 (Remarks to the Author):

In this report, Noh and colleagues successfully employ a novel approach to identify the first genome-wide significant finding for OCD (i.e. a rare coding variant burden in NRXN1, $p=6.37 \times 10^{-11}$). Historically, this is a very significant accomplishment, because no other findings in OCD genetics have reached this level of significance despite >100 candidate gene studies (meta-analyzed by Taylor et al, PMID 22665263), a dozen linkage studies (reviewed by Pauls et al, PMID 24840803) and two genome-wide association studies (PMIDs 22889921 and 24821223). The authors may wish to emphasize this point. The authors also introduce an innovative method that both integrates cross-species (mouse/dog/human) genetic data and provides a cost- and analytically-efficient approach to examine conserved non-coding elements. It is impressive to see this work on a behavioral trait and it will be very interesting to see if this method extends to other phenotypes (behavioral and otherwise).

We are grateful to the reviewer for acknowledging the novelty and significance of our findings. In accordance to the reviewer's comment, in our revised manuscript we emphasize the novelty and significance of our results in the context of current literature (page 15).

Strengths:

- 1. Despite a somewhat complicated, multi-step approach, the paper remains easy to follow and the results are clear.*
- 2. Rigor - the authors performed careful validation of pooled genotype findings (which can often cause problems) and included replication within this study as well as relating coding variants to the broader literature through ExAC.*
- 3. Despite using a candidate gene approach and targeted resequencing, the authors provided a strong justification for the genes (and associated gene regions) included.*
- 4. Overall results from Figure 1c, demonstrating that this candidate gene set is enriched for meaningful associations.*
- 5. Functional verification of regulatory variants via EMSA.*

We thank the reviewer for recognizing the strengths of our study. We indeed put significant efforts into providing clear description of our multi-step approach and performing rigorous statistical analysis. As the reviewer points out, we also carefully selected candidate genes that

are likely to be enriched with OCD-associated genes, which is confirmed in the study. In addition, we experimentally show that our candidate variants change transcription factor binding, which provides functional evidence beyond statistical association.

Weaknesses:

1. Many of the coding variants associated with OCD are synonymous. The authors should comment on how they envision these impacting gene function.

In response to the reviewer's comments, in our revised manuscript we describe how our synonymous candidate variants could have an impact on gene function (page 19 in main text and page 7 in supplementary information). Specifically, we suggest that synonymous variants could potentially affect yields of correctly folded proteins by disrupting RNA processing and post-transcriptional regulation or altering mRNA degradation (PMID 21878961). We also included further analysis where we found that all five of our synonymous candidate variants in *NRXN1* reside at the bases that are unusually fast-evolving (chr2:50,463,984, GERP++ -11.4; chr2:50,464,065 GERP++ -10.8, chr2:50,723,068 GERP++ -11.2) or slow-evolving (chr2:50,733,745, GERP++ 3.41; chr2:50,850,686, GERP++ 3.51), suggesting that these variants may have potential regulatory functions.

Reviewer #2 (Remarks to the Author):

This study used targeted sequencing and a new analysis method to identify genes and individual variants associated with OCD risk, and validate the result in independent data and by experiments, which may extend the power to elucidate the biological basis of OCD.

My concerns are:

1. How to select candidate genes for targeted sequencing is very important in this paper, but it's not clear from author's description: 1) "56 genes from regions associated in our GWAS of canine CD", how to define the regions associated in GWAS, the author gave two citations but how big the region is and why chose that length should be provided; 2) for ASD genes, I don't know which "whole genome study of ASD in humans" the author used (no citation was provided), the sample size and statistical power are big issues to consider; 3) for the human candidate genes, why author selected SCZ genes found in 2008 and 2009, but not included recent progress by PGC in past three years?

In response to the reviewer's comments, we now provide a more detailed explanation of our gene selection approach in our revised supplementary information (page 1). Below, we briefly summarize this additional information and explain why certain genes could not be included in the study:

- 1) Canine CD-associated regions were defined using linkage disequilibrium-based clumping around SNPs with $P < 0.0001$ (that is, SNPs within 1Mb with $r^2 > 0.8$ and $P < 0.01$). These thresholds were chosen because the canine GWAS was performed

within a single dog breed (Doberman pinscher), and dog breeds have extensive linkage disequilibrium and large haplotype blocks (~500kb-1Mb)(PMID: 16341006).

- 2) The ASD gene list was compiled from the evolving SFARI gene database (<https://gene.sfari.org/>) in 2009, which provides a filtered, annotated reference set. The references used by SFARI include research in animal models and human genetic studies. In our revised manuscript, we have provided a link to SFARI, which includes, for all ASD genes, their associated primary references, and supporting evidence for each gene. This would include the citation for the whole genome study in which the gene was discovered. We also updated our text in the revised manuscript (page 6).

A negligible (<0.2%) number of variants in SFARI are derived from GWAS, which require very large sample sizes for statistical power. Almost 90% of the annotated genetic variants are rare, damaging, highly penetrant variants identified using family studies (i.e. PMID 17363630).

- 3) We agree with the reviewer that it will be useful to consider results from PGC in applying our new method to future studies of psychiatric disease. However, when we designed the capture sequencing array for this study, the PGC results were not available, and thus could not be included in our study. Including these newly discovered genes in future sequencing studies could potentially identify new genes associated with OCD.

Despite potential limitations in the gene set included here (based on previous studies), our targeted approach was able to identify meaningful associations, suggesting its potential application in other complex diseases that have yet to achieve significant results from GWAS or other large-scale approaches.

2. Gene length is a confounding factor when counting the number of SNPs, longer genes tend to carry more SNPs theoretically. But no clear adjustment was provided in this study, which will produce bias result. Same problem exists when we compare the regulation regions of those genes.

We agree with the reviewer that it is imperative to account for gene length. Our method controls for gene length through a permutation that models that larger SNP counts expected for longer genes. Inspired by the reviewer's comment, we have included in our revised supplemental information (page 4) an assessment of how well this approach control for gene length. A linear regression of the gene-based p-values and the associated gene lengths results in a slope that is close to 0 (slope = -5.54×10^{-10} , intercept=0.7). We are therefore confident that our current approach successfully controlled for a potential bias introduced by gene length. We updated our manuscript to incorporate this new analysis (page 8).

3. The variant categories in PolyStrat were really confusing: what is the difference between “(i) all” and “(iv) ‘evolutionary’ (Evo) combining (ii) and (iii)” in evolution categories? The criteria of Cons and Div should be provided.

In response to the reviewer’s comment, in our revised manuscript we provide additional explanation to distinguish the categories between ‘all’ and ‘evolutionary’ (page 8). Briefly, the ‘All’ category contains all detected variants regardless their evolutionary status, whereas ‘Evo’ category contains variants that are annotated as ‘conserved’ or ‘divergent’. As a result, ‘All’ category contains 89% more variants than ‘Evo’ category (41,504 variants in ‘All’ vs. 22,006 variants in ‘Evo’).

4. In the EMSA validation section, if author proposed “incorporating both DHS and conservation” can better predict functional regulatory variants, they should construct a statistical model and provide more convincing data and analysis, such as the sensitivity and specificity results.

We agree that this is an important next step to follow up on our observation. Our current dataset of 17 SNPs is too small to construct a meaningful statistical model, and we had intended to note this EMSA result as an interesting observation, rather than a conclusion. In response to the reviewer’s comment, the revised manuscript more clearly states that our observation is consistent with previously published results that demonstrate the benefit of incorporating DHS and conservation in identifying regulatory variants (page 13-14).

5. Striatum is an important region linked to OCD, which author mentioned in summary part, but whether the strongly associated genes they found active in this region is not provided.

In response to the reviewer’s comment, in the revised manuscript and supplementary information we include the expression levels of the four genes in the striatum (page 19, manuscript; page 7, supplementary information). Specifically, from the Allen Brain Atlas’s microarray data (<http://human.brain-map.org/>), we extracted 990 \log_2 expression values of the 33 probes for the four genes, which are measured from 5 sub-regions of the striatum in 6 human individuals, and confirmed that all four genes are abundantly expressed in the human striatum (*NRXN1*, mean expression level[\log_2]=7.6 from 4 probes in 30 regions; *REEP3*, mean expression level[\log_2]=7.9 from 2 probes in 30 regions; *CTTNBP2*, mean expression level[\log_2]=8.3 from 2 probes in 30 regions; *HTR2A*, mean expression level[\log_2]=4.4 from 25 probes in 30 regions). These findings suggest that all of the four genes we identify here by PolyStrat are active in the striatum, further supporting their relevance to OCD.

6. The ExAC database is now updated to gnomAD (<http://gnomad.broadinstitute.org>), with more than double sample sizes and data. Comprehensive coding variants’ validation should use gnomAD database, especially for rare variants.

GnomeAD is an exciting, but very new, genomic resource. It only became available after the submission of this paper. Redoing the analysis using gnomAD would not be trivial, due to the

necessity of ensuring that the results are not influenced by population stratification, different sequencing platforms, gene length and number of transcripts. Our current gene-based analysis includes already public data for 33,370 control individuals, achieving highly significant association ($p=6.37 \times 10^{-11}$), and we believe that this provides sufficient evidence to the validity of our approach.

7. Author claimed this method can “highlighting new potential new therapeutic targets” in introduction, but to be precise, the targets are not new when using capture sequencing. This method can validate existing candidate genes, and explore new variants in coding and regulatory regions.

As reviewer #1 pointed out, despite >100 candidate gene studies (meta-analyzed by Taylor et al, PMID 22665263), a dozen linkage studies (reviewed by Pauls et al, PMID 24840803) and two genome-wide association studies in OCD (PMIDs 22889921 and 24821223), no other findings have reached the level of significance achieved in this study.

We used a very broad set of criteria to select 608 genes, or ~3% of all human genes. In our study, most of the “candidate genes” have no previously determined involvement in human OCD, but instead are chosen by criteria, such as association with related neuropsychiatric diseases or patterns of gene expression in the striatum. Our research is the first to conclusively link any of these genes to OCD, and in addition we implicate specific functional variants. We believe that, given none of these genes or variants were confidently associated with OCD prior to this study, they represent new potential therapeutic targets for OCD that would not have been previously considered. *Note: We corrected the typo (double usage of ‘new’) in the sentence (page 23).*

Point by point response to comments from the two reviewers.

We would like to thank the reviewers for taking the time to review our manuscript again. Both reviewers commented, in their second reviews, that our revised manuscript addressed all their concerns.

Reviewer #1 (Remarks to the Author):

The authors have addressed my concerns and I view the current version as acceptable for publication.

Reviewer #2 (Remarks to the Author):

The authors have addressed all my comments, and the answers are acceptable.